# Sodium Butyrate Induces Mitophagy and Apoptosis of Bovine Skeletal Muscle Satellite Cells through the Mammalian Target of Rapamycin Signaling Pathway

**DOI:** 10.3390/ijms241713474

**Published:** 2023-08-30

**Authors:** Yanling Ding, Pengfei Wang, Chenglong Li, Yanfeng Zhang, Chaoyun Yang, Xiaonan Zhou, Xiaowei Wang, Zonghua Su, Wenxuan Ming, Ling Zeng, Yuangang Shi, Cong-Jun Li, Xiaolong Kang

**Affiliations:** 1Key Laboratory of Ruminant Molecular and Cellular Breeding, College of Animal Science and Technology, Ningxia University, Yinchuan 750021, China; dingyanling950816@163.com (Y.D.); wpf960225@126.com (P.W.); chaoyuny@yeah.net (C.Y.);; 2Animal Genomics and Improvement Laboratory, Henry A. Wallace Beltsville Agricultural Research Center, Agricultural Research Service, United States Department of Agriculture USDA, Beltsville, MD 20705, USA

**Keywords:** bovine skeletal muscle satellite cells, sodium butyrate, mitophagy, apoptosis, mTOR signaling pathway

## Abstract

Sodium butyrate (NaB) is one of the short-chain fatty acids and is notably produced in large amounts from dietary fiber in the gut. Recent evidence suggests that NaB induces cell proliferation and apoptosis. Skeletal muscle is rich in plenty of mitochondrial. However, it is unclear how NaB acts on host muscle cells and whether it is involved in mitochondria-related functions in myocytes. The present study aimed to investigate the role of NaB treatment on the proliferation, apoptosis, and mitophagy of bovine skeletal muscle satellite cells (BSCs). The results showed that NaB inhibited proliferation, promoted apoptosis of BSCs, and promoted mitophagy in a time- and dose-dependent manner in BSCs. In addition, 1 mM NaB increased the mitochondrial ROS level, decreased the mitochondrial membrane potential (MMP), increased the number of autophagic vesicles in mitochondria, and increased the mitochondrial DNA (mtDNA) and ATP level. The effects of the mTOR pathway on BSCs were investigated. The results showed that 1 mM NaB inhibited the mRNA and protein expression of mTOR and genes AKT1, FOXO1, and EIF4EBP1 in the mTOR signaling pathway. In contrast, the addition of PP242, an inhibitor of the mTOR signaling pathway also inhibited mRNA and protein expression levels of mTOR, AKT1, FOXO1, and EIF4EBP1 and promoted mitophagy and apoptosis, which were consistent with the effect of NaB treatment. NaB might promote mitophagy and apoptosis in BSCs by inhibiting the mTOR signaling pathway. Our results would expand the knowledge of sodium butyrate on bovine skeletal muscle cell state and mitochondrial function.

## 1. Introduction

Sodium butyrate (NaB) is the sodium salt of short-chain fatty acids. It is a byproduct of dietary fiber produced during microbial fermentation in the gastrointestinal tract. NaB can cause cell cycle arrest [1], inhibit the growth of cancer cells [2], and regulate energy metabolism and mitochondrial function [3]. NaB inhibits angiogenesis, immune response, and oxidative stress; activates apoptosis; and even regulates the expression of non-coding RNAs in malignant cells [4]. NaB could inhibit the viability and significantly enhance the apoptosis of Hela cells in a time- and dose-dependent manner [5]. NaB also inhibits proliferation, stimulates apoptosis, and promotes differentiation of human colon cancer cells [6]. NaB reduces Ang II-induced NLRP3 inflammasome-related protein expression and repairs autophagy dysfunction due to lysosomal damage in macrophages [7]. For cattle, NaB induces profound changes in gene expression in bovine kidney epithelial cells, which were associated with cell cycle control [8,9]. In addition, NaB regulates chromatin state changes in bovine rumen epithelial primary cells [10,11] and promotes milk fat synthesis in bovine mammary epithelial cells [12]. Furthermore, NaB also plays an essential role in bovine immune and inflammatory responses [13]. Supplementation of NaB improved gastrointestinal health by regulating the bacterial populations in dairy calves [14]. The above studies showed the pleiotropic effects of NaB on biological processes in host cells.

Mitochondria are vital organelles for energy production and cellular sources of active substances, which play an essential role in maintaining energy homeostasis, cell growth, and development. Mitochondria also maintain intracellular calcium homeostasis in the organism. Skeletal muscle accounts for approximately 40% of the total body weight of mammal animals [15]. It is the main organ for maintaining the energy metabolism and locomotion of the organism, as well as the main site of glucose uptake. Thus, skeletal muscle plays a vital role in the glucose metabolism of the organism, which means that intracellular mitochondria in muscle cells are essential for maintaining muscle function [16]. Reactive oxygen species (ROS) are oxidants that cause the peroxidation of unsaturated fatty acids in lipids, and the organism producing plenty of ROS will damage intracellular lipids, proteins, and mitochondrial DNA (mtDNA), and changes in mtDNA levels will lead to mitochondrial dysfunction and reduced biosynthesis, which in turn will affect cell growth and metabolism [17].

The mammalian target of rapamycin (mTOR) is a member of the protein family related to a phosphatidylinositol kinase-related kinase (PIKK), which can integrate various extracellular signals such as nutrition, energy, and cytokines. mTOR is involved in transcription, protein translation, ribosomal synthesis, and other processes. Therefore, mTOR plays an important role in cell growth, apoptosis, and the metabolic process of the organism [18,19]. When the mTOR signaling pathway was inhibited, the cell metabolism reacted abnormally, which caused the results of slower cell growth and faster catabolism, as well as promoting apoptosis and mitophagy [20]. Studies have shown that the addition of NaB to skeletal muscle can induce mitochondrial function by stimulating energy expenditure in mice [21]; when mitochondria are damaged, overproduction of NADH and deficiency of NAD^+^ lead to an imbalance of redox status, enhanced oxidative stress, decreased mitochondrial membrane potential(MMP), increased ROS production, and abnormal ATP synthesis [22]. NaB improves colitis by inhibiting oxidative stress and NF-κB/NLRP3 activation in mice, which may be via COX-2/Nrf2/HO-1 activation and mitophagy [23]. Butyrate treatment could alleviate oxidative stress by regulating NRF2 nuclear accumulation and histone acetylation via GPR109A in bovine mammary epithelial cells [24]. It has been reported that NaB regulates breast cancer cell growth and induces apoptosis by activating the activity of caspase3 and caspase8, increasing intracellular ROS levels, decreasing MMP, and inducing cell cycle arrest [25]. The mechanism of NaB-induced apoptosis of human colon cancer cells was that NaB acts on mitochondria, causes changes in mitochondrial MMP and caspase cascade activation, dissolves proteins in cells or stimulates caspase to activate DNA enzyme to cause degradation of nuclear DNA, and finally leads to apoptosis [26].

Skeletal muscle is an important organ of energy metabolism and contains large numbers of mitochondria; NaB has multiple roles in regulating the cell state and gene expression of tissue from the host cell. However, it is unclear how NaB is involved in muscle cell-related activities and whether it affects mitochondria in myocytes. Therefore, this experiment was conducted to investigate the effects of NaB on apoptosis, mitophagy, and regulatory function of host skeletal muscle cells. This study will expand the knowledge of NaB and increase the understanding of the biological role of NaB on the mitochondrial function and energy utilization of skeletal muscle cells in livestock.

## 2. Results

### 2.1. NaB Inhibits BSC Proliferation and Promotes Apoptosis

The effects of 1 mM NaB on the proliferation of BSCs were detected by RT-qPCR and CCK-8. The results showed that compared to the control group, after 3 h, 6 h, and 12 h of 1 mM NaB treatment, the mRNA expressions of *CCNB2*, *CDK1*, and *PCNA* were significantly downregulated (*p* < 0.01). The inhibitory effect on BSCs proliferation was negatively correlated with time from 0 h to 12 h (Figure 1a). Cell viability was detected by the CCK-8 assay kit. The result showed that after treatment with 1 mM NaB for 12 h, 24 h, and 48 h, the cell viability significantly decreased (*p* < 0.01) (Figure 1b). For the effect of NaB on BSC apoptosis, the mRNA expression levels of pro-apoptosis genes *Bax* and *p53* were significantly upregulated at 12 h (*p* < 0.01), and the mRNA expression level of anti-apoptotic gene *Bcl-2* was downregulated significantly at 3 h–12 h after 1 mM NaB treatment (*p* < 0.01). These results indicated that 1 mM NaB promotes apoptosis of BSCs, and positively correlates with treating time from 0 h to 12 h (12 h was the most significant) (Figure 1c). Therefore, treating BSCs with 1 mM NaB for 12 h was preliminarily determined as the best condition. Flow cytometry assay was used to detect cell apoptosis. The results showed that compared with the control group, after 12 h treatment with 1 mM NaB, early apoptotic cells were increased, while late apoptotic cells and the rate of total apoptotic cells were significantly increased (*p* < 0.01) (Figure 1d). These results demonstrated that NaB inhibited the proliferation and promoted apoptosis of BSCs in a time-dependent manner.

### 2.2. NaB Promotes Mitophagy in BSCs

Studies have demonstrated that NaB induces mitophagy [24]. To clarify the effect of NaB on mitophagy in BSCs, RT-qPCR and WB assays were used to examine the influence of NaB on mitophagy at different concentrations and times. The results showed that compared to the control group, after NaB treatment of BSCs for 12 h at 0.5 mM, 1 mM, 2 mM, and 5 mM, the mRNA expression levels of pro-mitophagy genes *LC3B* and *PINK1* were significantly upregulated (*p* < 0.01), and anti-mitophagy gene *p62* expression level was significantly downregulated (*p* < 0.01), while the most significant change was at 1 mM NaB concentration (Figure 2a). The protein expression trend was consistent with the corresponding mRNA expression after 0.5 mM, 1 mM, 2 mM, and 5 mM NaB treatment at 12 h (Figure 2b). The above studies indicated that NaB promoted mitophagy in BSCs, and the most significant effect was treatment with 1 mM concentration NaB. To investigate the effect of NaB treatment on mitophagy at different time points, BSCs were treated with 1 mM NaB for 3 h, 6 h, and 12 h, and it was found that the expression levels of *LC3B* and *PINK1* were increased significantly with time(*p* < 0.01), and *p62* expression levels were decreased significantly (*p* < 0.01). Still, the degree of this downregulation gradually diminishes with treating time, and mitophagy was positively correlated with time from 0 h to 12 h, and most significantly at 12 h (Figure 2c). This indicated that NaB promotes mitophagy in a time- and dose-dependent manner. Based on the above experiments, cells were treated with 1 mM NaB for 12 h in subsequent experiments.

### 2.3. NaB Induces ROS/MMP Response in Mitochondria

Mitochondrial ROS and mitophagy were key factors in regulating cellular homeostasis, and a decrease in MMP levels was considered one of the hallmarks of mitophagy and apoptosis [27]. Therefore, after NaB-induced mitophagy, this study explored the changes in the levels of ROS and MMP. The results showed that after 1 mM NaB treatment, the proportion of cells with decreased MMP increased significantly (*p* < 0.01) (Figure 3a). ROS levels were detected by ROS assay kit. The result showed that compared to the negative control group, the ROS level in the positive control group was higher, which was approximately 3.5 times higher, while the ROS level was significantly higher than the blank control after 1 mM NaB treatment (*p* < 0.01) (Figure 3b). Laser confocal assay showed that the ROS levels were not significant in both the blank control and negative control groups, and the NaB treatment group was significantly higher than that in the negative control group (Appendix A). The present studies showed that NaB promotes mitophagy, but the morphological and structural changes of mitochondria needed to be clarified. To elucidate the morphological and structural changes of mitochondria after 1 mM NaB treatment cells at 12 h, the ultrastructure of mitochondria was observed using transmission electron microscopy. The results showed that compared to the control, NaB treatment increased the number of mitochondrial vacuolization and autophagosomes (Figure 3c). The above experiments showed that NaB induced the decrease in MMP, and increased the ROS level and the number of autophagosomes, which further verified the previous molecular results and suggested that NaB induced mitophagy.

### 2.4. NaB Promotes mtDNA Copy Number

mtDNA replication and transcription levels contribute to mitochondrial biogenesis, energy metabolism, and antioxidant capacity field [28]. This study investigated the effect of NaB on mtDNA copy number. The results showed that NaB treatment significantly upregulated the mRNA expression levels of ND1 and COX1, which were marker genes of mtDNA copy number, indicating an increase in intracellular mitochondria (Figure 4a). The mitochondria were the primary source of ATP production, and a large amount of ATP production accompanied the increase in the number of mitochondria, so the test further verified the changes in intracellular ATP. The result found that the content of intracellular ATP increased significantly after NaB treatment (*p* < 0.01) (Figure 4b), which was consistent with the expected results. These results indicated that 1 mM NaB treatment increased the number and activity of intracellular mitochondria. The increased activity of mitochondria might produce a large amount of ATP to maintain the energy metabolism of the organism.

### 2.5. NaB-Mediated mTOR Signaling Pathway Promotes Mitophagy and Apoptosis

Increased mitochondrial ROS levels and MMP deficiency were common phenomena during mitophagy-mediated cell apoptosis [29]. Numerous studies have shown that the mTOR signaling pathway mediated mitophagy and apoptosis in colon, ovarian [30], and other cancer cells [31,32]. To investigate the potential mechanisms of NaB on mitophagy and apoptosis in BSCs, this study detected the changes in the mTOR signaling pathway after 1 mM NaB treatment. The results showed that compared to the control group, after 1 mM NaB treatment of BSCs, the mRNA expression levels of critical genes *mTOR*, *EIF4EBP1*, and *AKT1* in the mTOR signaling pathway were highly significantly downregulated (*p* < 0.01), and *FOXO1* was significantly downregulated (*p* < 0.05) (Figure 5a). PP242 was an effective inhibitor of the mTOR signaling pathway. PP242 inhibitor with different concentrations (50 nM, 100 nM, 200 nM) was used to treat BSCs. It was found that compared to the control group, after treatment with PP242 inhibitors at 50 nM, 100 nM, and 200 nM, the mRNA expression levels of *mTOR*, *EIF4EBP1*, *FOXO1*, and *AKT1* in the mTOR signaling pathway were significantly downregulated (*p* < 0.01), and most significantly at 100 nM (Figure 5b). After treatment of BSCs with 1 mM NaB and 100 nM PP242 inhibitor, respectively, the protein expression levels of mTOR, EIF4EBP1, FOXO1, and AKT1 in the mTOR pathway were consistent with the mRNA levels, which also downregulated significantly (*p* < 0.01) (Figure 5c). Moreover, in the 100 nM PP242 treatment BSCs, the expression levels of the pro-apoptotic genes *Bax* and *p53* were significantly upregulated (*p* < 0.01), and the anti-apoptotic genes *Bcl-2* were significantly downregulated (*p* < 0.01) (Figure 5d). The expression levels of pro-mitophagy genes *LC3B* and *PINK1* were significantly upregulated (*p* < 0.01), and anti-mitophagy *p62* was significantly downregulated (*p* < 0.01) (Figure 5e). This indicated that PP242 could promote mitophagy and apoptosis after BSCs were treated by PP242 inhibitor, which could be speculated that PP242 inhibitor and NaB have the same effect on BSCs regulation. The above results showed that NaB might promote mitophagy and apoptosis in BSCs by inhibiting the mTOR signaling pathway.

## 3. Discussion

In different cancer cell lines, NaB can induce growth, differentiation, and apoptosis. NaB inhibits proliferation and induces apoptosis in human breast cancer cells in a dose-dependent manner, thus being an effective inhibitor of breast cancer [33]. NaB also inhibits nasopharyngeal carcinoma cells and promotes apoptosis by regulating the progression of the G0/G1 phase of the cell cycle and activating the apoptotic factor field [34]. NaB alleviates the adaptive response to inflammation, modulating fatty acid metabolism in lipopolysaccharide-stimulated bovine hepatocytes [35]. In addition, NaB promotes milk fat synthesis in bovine mammary epithelial cells via the *GPR41* gene and its downstream signaling pathways [12]. NaB reduces the excessive formation of neutrophil extracellular traps by inhibiting autophagy, thus reducing the damaging effect of neutrophil extracellular traps on bovine mammary epithelial cells [36]. NaB affects rumen epithelial morphology and plasma concentrations of hormones by increasing plasma concentration of glucagon-like peptide-2 and lower gene expression of *TNF-α*, *IL-1β*, and *TLR-2* in the cow’s rumen epithelium cells [37]. NaB-induced lipid accumulation increased expression of peroxisome proliferator-activated receptor gamma (PPARγ) by enriching H3 acetylated K27 (H3K27), thereby affecting growth performance, lipid metabolism, and fatty acid composition in the skeletal muscle of offspring piglets [38]. Moreover, NaB inhibited platelet-derived growth factor-induced proliferation and migration in pulmonary artery smooth muscle cells through Akt inhibition [39].

NaB could modulate the muscle and adipose tissue metabolism at the transcriptional level by decreasing the amino acid metabolism pathways of growing pigs [40]. However, the effect of NaB on the proliferation and apoptosis of BSCs and the mechanism of action are not apparent. Therefore, this experiment investigated the effect of 1 mM NaB on the proliferation and apoptosis of BSCs. The results showed that NaB significantly inhibited the mRNA expression levels of proliferating marker genes *CCNB2*, *CDK1*, *CDK2,* and *PCNA*. *Bcl-2* is a critical anti-apoptotic gene; *Bax* and *p53* genes play an activating role in apoptosis. In this study, the expression levels of the *Bcl-2* gene were significantly downregulated, while *Bax* and *p53* were significantly upregulated after 1 mM NaB treatment. The apoptosis rate was detected by flow cytometry. It was found that the total number of apoptotic cells was significantly increased after 1 mM NaB treatment. These results showed that NaB inhibited the proliferation and promoted apoptosis of BSCs. The study investigated NaB inhibiting U937 cell growth and apoptosis in a dose-dependent manner, mainly by upregulation of the expression of the pro-apoptotic gene *Bax* and downregulation of the anti-apoptotic gene *Bcl-2* [41]. We detected the effect of NaB on the proliferation and apoptosis of osteosarcoma cells and its regulatory mechanism. The results found that the protein expression level of Bcl-2 decreased while Bax increased after NaB treatment. In addition, p53 inhibited proliferation and promoted apoptosis, while MDM2 promoted proliferation and inhibited apoptosis. NaB enhanced p53 expression and decreased MDM2 expression, indicating that NaB promoted apoptosis by regulating the MDM2-p53 signaling pathway [42]. The above studies were consistent with our results, indicating that NaB promoted cell apoptosis.

Cell apoptosis was essential for tissue development, cellular homeostasis, and disease control, and it was usually activated by intrinsic mitochondria-dependent pathways and extrinsic cell apoptotic pathways [43]. This study found that NaB promoted apoptosis in BSCs, but the molecular mechanism causing apoptosis still needs clarification. Therefore, the test further explored the relationship between NaB-induced mitophagy and cell apoptosis. The results showed that the mRNA and protein expression levels of pro-mitophagy genes LC3B and PINK1 were promoted. In contrast, the mRNA and protein expression levels of anti-mitophagy gene p62 were significantly downregulated after 0 mM, 0.5 mM, 1 mM, and 2 mM NaB treatment cells for 3 h, 6 h, and 12 h, which indicated that NaB promoted mitophagy. Mitophagy acts as a protective scavenger to maintain intracellular environmental homeostasis by digesting different products of the organism [44]. Many studies have shown that mitophagy affects the process of cell apoptosis [25,45], and excessive mitophagy exerts cytotoxic effects and activates the cell apoptotic process [46]. Thus, mitophagy can act as an activator of cell apoptosis. Mitophagy usually accompanies the production of ROS, damage to MMP, and increased mtDNA, where damage to MMP and increased mtDNA were markers of cell apoptosis. NaB promoted the expression of mitophagy marker proteins *LC3* and *PINK1*. It decreased MMP in hamster ovary cells in a time- and dose-dependent manner, while the expression levels of the apoptosis marker proteins PARP, Bcl-2, and caspase-3 gradually increased. The mechanism of NaB regulation of apoptosis in hamster ovary cells was due to the activation of caspase 9 by cytochrome C (Cyt C) in the mitochondrial electron transport chain, and the activation of caspase 9 led to the activity of downstream effectors caspase 3 and Bcl-2, which eventually led to cell apoptosis [47]. In this study, NaB-induced mitophagy was followed by a decrease in MMP and significant upregulation of ROS levels, along with a large amount of mtDNA and ATP production by the cells. These results tentatively suggested that NaB-induced cell apoptosis was mediated by mitophagy. NaB induced estrogen receptor alpha (ERα) expression, activated the phosphorylation of AMPK and PGC1α, and improved mitochondrial aerobic respiration in cultured skeletal muscle cells in ovariectomized mice [48]. In addition, cisplatin combined with NaB increased the apoptosis and the intracellular ROS of gastric cancer cells, decreasing the MMP. Western blotting verified that the combination of sodium butyrate and cisplatin remarkably enhanced the levels of mitochondrial apoptosis-related pathway proteins [49]. The above studies were consistent with the present research in BSCs.

In this study, NaB promoted mitophagy and apoptosis in BSCs. The molecular mechanisms and whether downstream signaling pathways regulated them remain to be explored. mTOR proteins are highly conserved serine/threonine (Ser/Thr) protein kinases belonging to the PIKK protein family. mTOR exists as two protein complexes: mTORC1 and mTORC2, which regulate cellular metabolism, growth, mitophagy, and apoptosis by participating in multiple signaling pathways in vivo [50]. mTOR inhibitors block mTOR signaling pathways and produce anti-inflammatory, anti-proliferative, mitophagous, and apoptosis-inducing effects [51]. Based on the above functions of mTOR, we studied the mTOR signaling pathway using NaB treatment of BSCs. We found that NaB treatment caused significant downregulation of mRNA and protein expression levels of mTOR, EIF4EBP1, FOXO1, and AKT1 in the mTOR signaling pathway, which indicated that NaB inhibited the mTOR signaling pathway. The mTOR signaling pathway inhibitor PP242 also inhibited the expression levels of mTOR, EIF4EBP1, FOXO, and AKT1. In addition, the expression levels of apoptotic genes *Bax* and *p53* were significantly upregulated, and the expression levels of anti-apoptotic gene *Bcl-2* were significantly downregulated. In contrast, the expression levels of mitophagy genes *LC3B* and *PINK1* were significantly upregulated. Anti-mitophagy gene *p62* expression levels were significantly downregulated, suggesting that NaB has a synergistic effect with PP242 inhibitor, which promoted mitophagy in BSCs by inhibiting the mTOR signal pathway, thereby inducing cell apoptosis. In nasopharyngeal carcinoma cells, NaB induces mitophagy-related apoptosis by inhibiting the AKT/mTOR axis and then exerts anti-tumor effects in nasopharyngeal carcinoma [52]. Research showed that NaB inhibited bladder cancer cell migration and induced AMPK/mTOR pathway-activated mitophagy and ROS overproduction through the miR-139-5p/Bmi-1 axis, while ROS overproduction contributed to NaB-induced cysteine-dependent apoptosis [53]. α-synuclein aggregation is central to the pathogenesis of Parkinson’s disease. It was found that in murine neuroendocrine STC-1 cells, NaB leads to α-synaptic nuclear protein degradation and apoptosis through Atg5-dependent and PI3K/Akt/mTOR-related mitophagy pathways [54].

## 4. Materials and Methods

### 4.1. Reagents and Antibodies

Fetal bovine serum (FBS) and 0.25% Trypsin-EDTA were obtained from Gibco (Gibco, Grand Island, NY, USA). The whole protein extraction kit and BCA protein content assay kit were purchased from KeyGEN Biotechnology (KeyGEN, Nanjing, China). CCK-8 assay kit (APExBIO, Houston, TX, USA), mTOR pathway inhibitor PP242 (APExBIO, Houston, TX, USA), ROS assay kit (Elabscience Biotechnology, Wuhan, China), sodium butyrate (Sigma-Aldrich, St. Louis, MO, USA), and MMP assay kit was purchased from Beyotime (Shanghai, China). SDS-PAGE gel kit (Solarbio, Beijing, China), GAPDH antibody (Cambridge, UK), and P62 antibody were obtained from Abcam (Cambridge, UK). mTOR antibody, AKT1 antibody, and FOXO antibody were purchased from Proteintech (Proteintech, Wuhan, China).

### 4.2. Cell Culture

Muscle tissues from the hindlimbs of the 1-month-old bovine fetus were minced into a coarse slurry and serially digested using 2% collagenase I for 1 h and in trypsinase for 30 min at 37 °C. Cells were then filtered from debris, centrifuged, and cultured. For culturing, the skeletal muscle satellite cells were maintained in cell culture plates and passaged until 70% confluence was reached in DMEM-F12 complete medium supplemented with 20% FBS and antibiotics (1% penicillin-streptomycin); cells were washed twice with phosphate-buffered saline (PBS) (HyClone, Cambridge, MA, USA) and digested with trypsinase for 4–5 min. The digestion was terminated by adding medium immediately when the cells were observed to become rounded and suspended. The fresh cells were collected after centrifugation for use or frozen and stored in liquid nitrogen for the next experiment. All cells were grown at 37 °C in a humidified atmosphere of 95% air and 5% CO_2_. The results of cell culture status are shown in Appendix A.

### 4.3. Cell Viability Assay

BSCs were seeded in 96-well plates at a density of 1 × 10^4^ cells per well. The cells were cultured in the incubator for 0 h, 12 h, 24 h, and 48 h, respectively, with a final concentration of 1 mM NaB after the cells were attached to the wall and the density reached 20%. Then, 10 μL CCK-8 reagent was added to each well and incubated for 2 h at 37 °C, and absorbance at 450 nm was measured using a microplate reader (BioTek Instruments, Winooski, VT, USA).

### 4.4. Cell Apoptosis Assay

BSCs were inoculated with 1 × 10^6^ cells per well in 6-well plates. When the density reached 70%, the culture medium was discarded and washed twice with PBS buffer, after which 1 mM NaB was added and incubated in a CO_2_ incubator for 0 h, 3 h, 6 h, and 12 h, respectively. The effect of NaB on cell apoptosis was detected by RT-qPCR and flow cytometry (FACSAria II, Becton Dickinson, Franklin Lakes, NJ, USA). The RT-qPCR reaction system consisted of 2 × ChamQ Universal SYBR qPCR Master Mix 10 μL, primer F 0.8 μL, primer R 0.8 μL, cDNA template 2 μL, and RNase-free ddH_2_O 6.4 μL. The reaction procedure was as follows: predenaturation at 95 °C for 30 s, denaturation at 95 °C for 10 s, and annealing extension for 30 s, 40 cycles in total.

### 4.5. Assay of Mitophagy and ROS

Cells were treated with NaB at concentrations of 0.5 mM, 1 mM, 2 mM, and 5 mM for 0 h, 3 h, 6 h, and 12 h, respectively, and total cellular mRNA and total protein were extracted. The effects of NaB on mitophagy genes and proteins at different times and concentrations were detected by RT-qPCR and Western blotting (WB). ROS levels were detected by ROS assay kit and laser confocal method, according to the assay kit operation manual. Determination of unspecific ROS levels by ROS assay kit: dichloro-dihydro-fluorescein diacetate (DCFH-DA, 10 μM) was treated with BSC cells for 60 min in the dark at 37 °C. After rinsing three times with buffer solution, 0.25% Trypsin EDTA was used to digest cells, and a complete culture medium was added to terminate digestion. Cells were collected by centrifugation at 1000× *g* for 10 min; Finally, the collected cells were resuspended in buffer solution, and the optimal excitation wavelength of 500 nm and optimal emission wavelength of 525 nm were selected using a fluorescence enzyme-labeled instrument for fluorescence detection.

### 4.6. MMP Assay

MMP was assayed by JC-1 staining. Cells were treated with NaB at 1 mM for 12 h; cells were collected and incubated at 37 °C for 20 min with 5 mg/mL JC-1 dye; 500 μL PBS was added to wash the cells three times, and the cells were placed in a fresh serum-free medium. The level of MMP was detected by flow cytometry.

### 4.7. Mitophagy Vesicles Assay

Cells treated with 1 mM NaB were incubated for 12 h. The medium was discarded, and 2.5% glutaraldehyde fixative was added and fixed at 4 °C for 6 h, followed by 1% osmium tetroxide for 2 h at room temperature. Samples were dehydrated twice by gradients of 30%, 50%, 70%, 80%, 85%, 90%, 95%, and 100% ethanol for 20 min each time. The mitophagy vesicles in mitochondria were observed using transmission electron microscopy by permeabilization, embedding, ultrathin sectioning, and double staining.

### 4.8. mtDNA Copy Number

Genomic DNA was extracted from the cells using a commercially available kit following the manufacturer’s instructions, calculated standard curve, and the expression of mtDNA copy number gene ND1 and COX1 in BSCs by 1 mM NaB treatment were detected for RT-qPCR, GAPDH was used as an internal reference gene to calculate the relative expression of each gene. The gene sequences were obtained from NCBI, and the primers were designed using primer5.0 software (Premier Bosoft, Palo Alto, CA, USA). The sequences of all primers are listed in Appendix A.

### 4.9. ATP Content Assays

When the cell density reached 70%, the culture medium was discarded, and the cells were washed twice with PBS buffer. A total of 200 μL of cell lysate was added to each well, and the cells were repeatedly resuspended using a pipette to lyse the cells fully. After centrifuging at 12,000× *g* for 5 min at 4 °C, the supernatant was taken, and the content of ATP in the whole cell lysate was measured using the ATP assay kit (Beyotime Biotech, Shanghai, China), according to the assay kit operation manual.

### 4.10. Western Blot

The RIPA lysis buffer was used to separate proteins from BSCs. Then, SDS-PAGE was used to separate the protein sample and then transfer it to the PVDF membrane and 3% albumin from bovine serum (BSA) for sealing. Then, the membrane was incubated overnight at 4 °C with primary antibody against PINK1 (1:1000), P62 (1:1000), LC3B (1:800), mTOR (1:1000), FOXO1 (1:800), AKT (1:1000), and GAPDH (1:500). On the second day, the membrane was incubated with the corresponding goat anti-mouse IgG or goat anti-rabbit IgG secondary antibodies at room temperature for 1 h. After washing, membranes were detected by applying a chemiluminescent reagent and visualized using Image Lab software (Version 4.0) (Hercules, CA, Bio-Rad). Image J was employed for quantitative analysis.

### 4.11. Effect of mTOR Signaling Pathway on Mitophagy and Apoptosis

Cells were treated with 1 mM NaB for 12 h, and total cellular mRNA and total protein were extracted. The mRNA and protein expression levels of mTOR, EIF4EBP1, FOXO1, and AKT1 in the mTOR signaling pathway were detected by RT-qPCR and WB. To further investigate the effect of the mTOR signaling pathway on mitophagy and apoptosis, the mTOR signaling pathway inhibitor PP242 was cultured with final concentrations of 0 nM, 50 nM, 100 nM, and 200 nM for 12 h, respectively. Total cellular mRNA and total protein were extracted to detect the effect of PP242 inhibitor on the expression of key genes in the mTOR signaling pathway, such as mTOR, EIF4EBP1, FOXO1, and AKT1, as well as the effects on mitophagy and apoptosis.

### 4.12. Statistical Analysis

The experimental data were statistically normalized using the 2^−ΔΔCt^ method and analyzed using the SPSS 25.0 software. One-way ANOVA was used for test analysis, and GraphPad Prism 8.0 software was used for graphing. The results were expressed by mean ± standard error. *p* < 0.05 indicated a significant difference.

## 5. Conclusions

NaB is an essential nutritional element for ruminants with specific biological activities. Our study aimed to investigate the role of NaB treatment on the proliferation, apoptosis, and mitophagy of bovine skeletal muscle satellite cells (BSCs). We demonstrated that NaB inhibited proliferation and increased apoptosis in BSCs. Further, NaB treatment significantly promoted mitophagy, increased ROS concentration, and decreased the MMP. The results also show that NaB might regulate apoptosis and mitophagy in BSCs by downregulating the mTOR signaling pathway. These findings provide novel insights into the mechanisms underlying how NaB regulates apoptosis and mitophagy, and indicate that NaB may be a critical metabolic molecule in muscle development, energy metabolism, and the regulation of mitochondrial function.

## Figures and Tables

**Figure 1 ijms-24-13474-f001:**
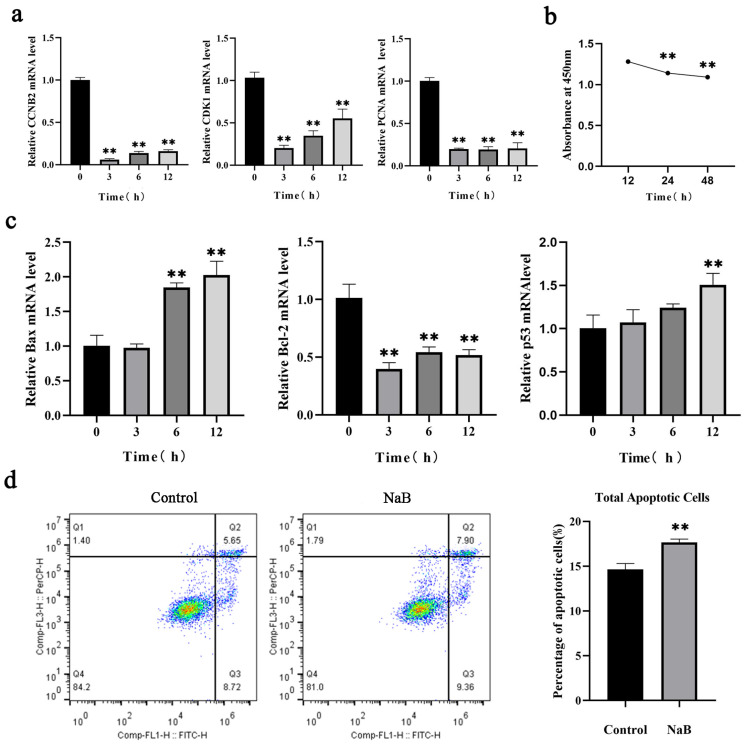
Effect of 1 mM NaB on the proliferation and apoptosis of BSCs. (**a**) The mRNA expression of proliferation genes *CCNB2*, *CDK1*, and *PCNA* with 1 mM NaB treatment. (**b**) The effect of 1 mM NaB treatment on cell viability. (**c**) The mRNA expression of apoptosis genes *Bax*, *Bcl-2*, and *p53* with 1 mM NaB treatment. (**d**) The apoptosis rate of cells with 1 mM NaB treatment by flow cytometry. Data are expressed as the mean ± standard deviation of three replicate experiments, ** *p* < 0.01.

**Figure 2 ijms-24-13474-f002:**
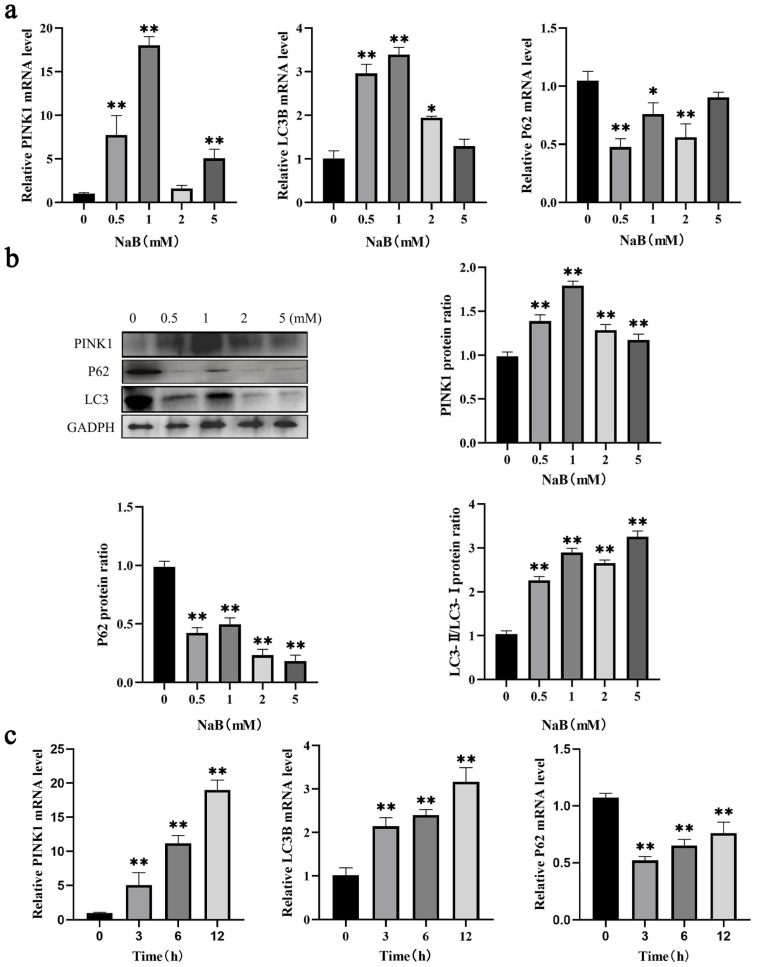
Effect of NaB on mitophagy in BSCs. (**a**) Expression of mitophagy genes *PINK*, *LC3B,* and *P62* with different concentrations of NaB treatment. (**b**) Expression of mitophagy proteins PINK, LC3B, and P62 with varying concentrations of NaB treatment. (**c**) Expression of mitophagy genes *PINK*, *LC3B*, and *P62* with different times of NaB treatment. * *p* < 0.05, ** *p* < 0.01.

**Figure 3 ijms-24-13474-f003:**
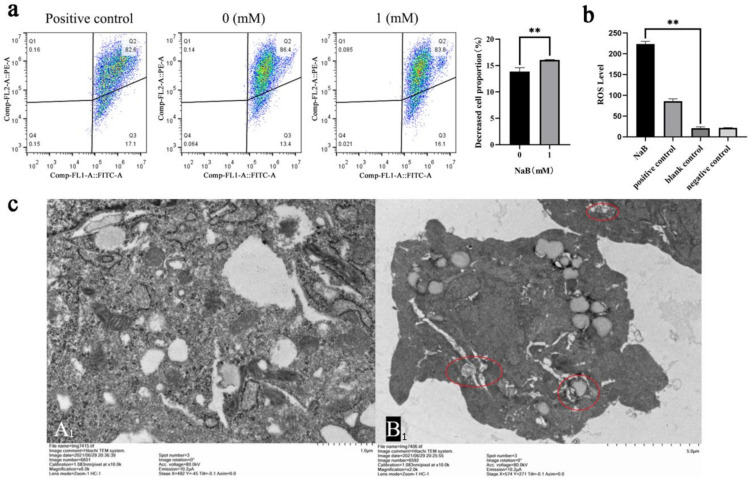
Effect of NaB on ROS and MMP in BSCs. (**a**) The percentage of cells with decreased MMP. (**b**) The impact of 1 mM NaB on cellular ROS level. (**c**) Ultrastructure of mitochondria under a transmission electron microscope (TEM). ((**A1**): Control group; (**B1**): sodium butyrate treatment group. The red circle in B represents autophagosomes). ** *p* < 0.01.

**Figure 4 ijms-24-13474-f004:**
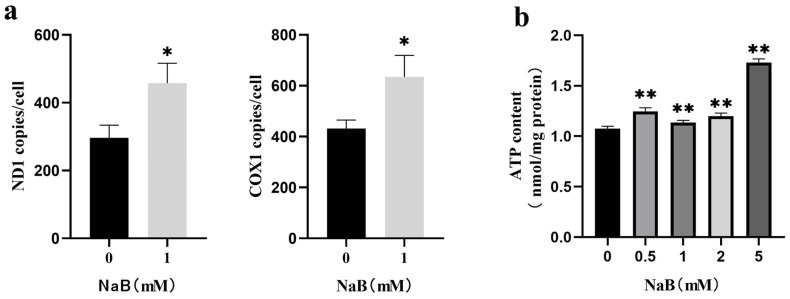
Effect of NaB on mtDNA and ATP content. (**a**) The expression of mtDNA copy number genes *ND1* and *COX1* with 1 mM NaB treatment. (**b**) Changes in cellular ATP content with different concentrations of NaB treatment. * *p* < 0.05, ** *p* < 0.01.

**Figure 5 ijms-24-13474-f005:**
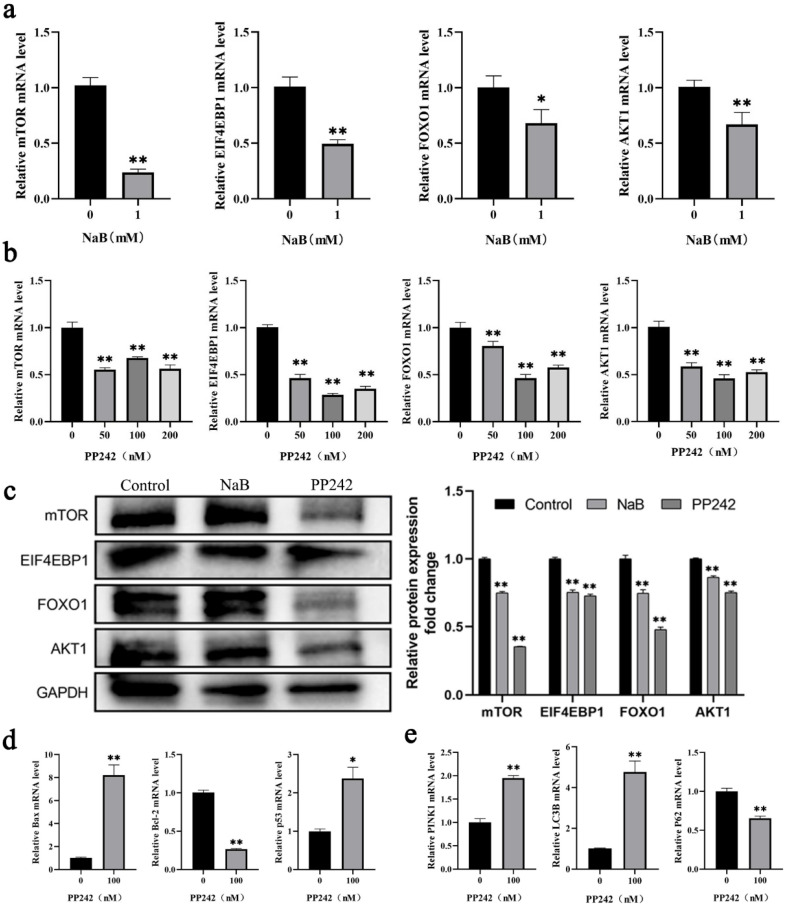
Effect of NaB on mitophagy and apoptosis through the mTOR signaling pathway. (**a**) The mRNA expression of key genes *mTOR*, *EIF4EBP1*, *FOXO1*, and *AKT1* in the mTOR signaling pathway with NaB treatment. (**b**) The mRNA expression of key genes *mTOR*, *EIF4EBP1*, *FOXO1*, and *AKT1* in the mTOR signaling pathway with the PP242 treatment. (**c**) The expression of key proteins mTOR, EIF4EBP1, FOXO1, and AKT1 in the mTOR signaling pathway with NaB and PP242 treatment, respectively. (**d**) The mRNA expression of apoptosis genes *Bax*, *Bcl-2*, and *p53* with the PP242 treatment. (**e**) The mRNA expression of mitophagy genes *PINK1*, *LC3B*, and *P62* with the PP242 treatment. * *p* < 0.05, ** *p* < 0.01.

## Data Availability

All data are available, and the correspondent can be contacted if requested.

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
