# Peer review of "Sodium Butyrate Induces Mitophagy and Apoptosis of Bovine Skeletal Muscle Satellite Cells through the Mammalian Target of Rapamycin Signaling Pathway"

_ijms, 2023, doi:10.3390/ijms241713474_

Round 1
Reviewer 1 Report
IJMS-2540070 Peer-Review Report: Sodium butyrate induces mitophagy and apoptosis of bovine skeletal muscle satellite cells through the mTOR signaling pathway
Summary/ General concept comments
Generally, the manuscript is generally clear, relevant to the field and presented in a well-structured manner and backed with recent publications cited. The study adds to the knowledge of mitophagy induction and apoptosis by sodium butyrate of bovine skeletal muscle satellite cells via the mTOR signaling route. However, there are concerns that the authors need to address to move the review process forward.
I found many sentences in the manuscript abnormally long, thus rendering them too hard to follow or understand. Enough time should be devoted to breaking these sentences into multiple sentences so readers can understand. For instance, lines 22-27 should be broken down into three sentences: The effects of the mTOR pathway on BSCs were investigated. The results showed that 1mM NaB inhibited the mRNA and protein expression of mTOR and genes AKT1, FOXO1, and EIF4EBP1 in the mTOR signaling pathway. In contrast, the addition of PP242, an inhibitor of the mTOR signaling pathway, also inhibited mRNA and protein expression levels of mTOR, AKT1, FOXO1, and EIF4EBP1, and promoted mitophagy and apoptosis, which were consistent with the effect of NaB treatment.
In addition to many other sentences, the following MUST be broken down into 2 -4 sentences. These are Lines 35-43, 43-48, 49-54, 63-67, 92-97, 98-103, 116-124, 139-153, 179-184, 188-195, 207-211, 212-219, 220-226, 226-232, 232-237, 292-299 and 303-310.
Specific comments
Abstract:
The abstract is well-written; however, it could still be improved. I have made a few observations for the authors to address.
Lines 14,36 The spelling of fiber is a non-British variant. For consistency, consider replacing it with the British English spelling.
Lines 27-28 The phrase ‘in conclusion’ should be deleted as it appears out of place here in an abstract.
Introduction:
The introduction is well-written with citations from relatively recent publications. It could still be spiced up a little with more recent publications.
Results
Line 139 The authors are missing the conjunction that before the word after. This should be added.
Discussion
Line 220 NaB induced is missing a hyphen; consider adding the hyphen.
Line 225 The verb modulated after could does not appear in the correct form; think of changing the verb form.
Line 293 The spelling of anti-tumor is a non-British variant. For consistency, consider replacing it with the British English spelling.
Line 298 The noun form of Parkinson is not correct; The authors should change it to its correct noun form.
Materials and methods
Line 336 RNase free is missing a hyphen; the authors should add the hyphen appropriately.
Line 343-44 According to the authors, ROS levels were detected by ROS assay kit and laser confocal method, according to the assay kit operation manual. It would be worthwhile to briefly describe the methodology so readers can follow it.
Line 362 Genes as it is in the sentence does not appear correct; the authors should consider changing it to singular.
Conclusions
This should be improved upon to reflect on the hypothesis and results obtained.
References
The MDPI guidelines on references were not strictly adhered to. The authors must make all the necessary amendments in all the cited references.

This has already been pointed out in the main report.
Author Response
Dear Editor,
Thank you so much for taking the time to review this manuscript. We are very grateful for all comments and suggestions from reviewers and editors, which would make this manuscript more reasonable and readable. Based on the latest comments from editors, all comments have been responded to correspondingly and language errors of this manuscript have been edited and polished completely and expertly. We really hope this revised version could address all your relevant comments. All comments and changes were shown in red in this new revised version.
Summary/ General concept comments
Generally, the manuscript is generally clear, relevant to the field and presented in a well-structured manner, and backed with recent publications cited. The study adds to the knowledge of mitophagy induction and apoptosis by sodium butyrate of bovine skeletal muscle satellite cells via the mTOR signaling route. However, there are concerns that the authors need to address to move the review process forward. I found many sentences in the manuscript abnormally long, thus rendering them too hard to follow or understand. Enough time should be devoted to breaking these sentences into multiple sentences so readers can understand. For instance:
Line 22-27 The effects of the mTOR pathway on BSCs were investigated, and the results showed that 1mM NaB inhibited the mRNA and protein expression of mTOR and genes AKT1, FOXO1, and EIF4EBP1 in the mTOR signaling pathway, while the addition of PP242, an inhibitor of the mTOR signaling pathway, also inhibited mRNA and protein expression levels of mTOR, AKT1, FOXO1, and EIF4EBP1, and promoted mitophagy and apoptosis, which were consistent with the effect of NaB treatment.
Response 1: The effects of the mTOR pathway on BSCs were investigated. The results showed that 1mM NaB inhibited the mRNA and protein expression of mTOR and genes AKT1, FOXO1, and EIF4EBP1 in the mTOR signaling pathway. In contrast, the addition of PP242, an inhibitor of the mTOR signaling pathway, also inhibited mRNA and protein expression levels of mTOR, AKT1, FOXO1, and EIF4EBP1, and promoted mitophagy and apoptosis, which were consistent with the effect of NaB treatment.
In addition to many other sentences, the following MUST be broken down into 2 -4
sentences. These are Lines 35-43, 43-48, 49-54, 63-67, 92-97, 98-103, 116-124, 139-
153, 179-184, 188-195, 207-211, 212-219, 220-226, 226-232, 232-237, 292-299 and
303-310.
Response 2: All these sentences have been broken down into several short sentences. We tried our best to improve the manuscript and made some changes to the manuscript. These changes will not influence the content and framework of the paper. And here we did not list the changes but marked in red in the revised paper.
Specific comments
Abstract:
The abstract is well-written; however, it could still be improved. I have made a few observations for the authors to address.
Lines 14,36 The spelling of fiber is a non-British variant. For consistency, consider replacing it with the British English spelling.
Response 3: Thank you for your reminder. The spelling of ‘fiber’ was replaced by ‘fibre’.
Lines 27-28 The phrase ‘in conclusion’ should be deleted as it appears out of place here in an abstract.
Response 4: Thank you so much. ‘in conclusion’ has been deleted in the manuscript.
Introduction:
The introduction is well-written with citations from relatively recent publications. It could still be spiced up a little with more recent publications.
Response 5: We sincerely appreciate the valuable comments. We have checked the literature carefully and added more references to the introduction part of the revised manuscript.
- Bai, X.; Wang, Y.; Liu, P.; Xia, W.; Wang, Y., Sodium butyrate regulation of NLRP3-Ser295 phosphorylation inhibits hypertensive nephropathy. Journal of Functional Foods 2023, 107, 105670.
- Ma, L.; Yang, Y.; Liu, W. H.; Bu, D. P., Sodium butyrate supplementation impacts the gastrointestinal bacteria of dairy calves before weaning. Appl Microbiol Biot 2023, 107, (10), 3291-3304.
- Bian, Z. B.; Zhang, Q. Y.; Qin, Y.; Sun, X. D.; Liu, L. L.; Liu, H. H.; Mao, L. Z.; Yan, Y. R.; Liao, W. Z.; Zha, L. Y.; Sun, S. X., Sodium Butyrate Inhibits Oxidative Stress and NF-kappa B/NLRP3 Activation in Dextran Sulfate Sodium Salt-Induced Colitis in Mice with Involvement of the Nrf2 Signaling Pathway and Mitophagy. Digest Dis Sci 2023, 68, (7), 2981-2996.
Results
Line 139 The authors are missing the conjunction that before the word after. This should be added.
Response 6: Thanks, we have revised the sentence. ‘that’ has been added before the word ‘after’ in the text.
Discussion
Line 220 NaB induced is missing a hyphen; consider adding the hyphen.
Response 7: Thanks for your correction. NaB induced has been replaced by “NaB-induced”.
Line 225 The verb modulated after could does not appear in the correct form; think of changing the verb form.
Response 8: Thanks a lot. The word modulated has been changed by ‘modulate’.
Line 293 The spelling of anti-tumor is a non-British variant. For consistency, consider replacing it with the British English spelling.
Response 9: Thank you. The spelling of anti-tumor has been replaced by anti-tumour.
Line 298 The noun form of Parkinson is not correct; The authors should change it to its correct noun form.
Response 10: Parkinson has been changed by ‘Parkinson’s’.
Materials and methods
Line 336 RNase free is missing a hyphen; the authors should add the hyphen appropriately.
Response 11: RNase free has been added a hyphen: RNase-free.
Line 343-44 According to the authors, ROS levels were detected by ROS assay kit and laser confocal method, according to the assay kit operation manual. It would be worthwhile to briefly describe the methodology so readers can follow it.
Response 12: Thanks for your suggestion. We have briefly described the method in section 4.5 of the article.
Determination of unspecific ROS levels by ROS assay kit: dichloro-dihydro-fluorescein diacetate (DCFH-DA, 10 μM) was treated with BSCs cells for 60 min in the dark at 37°C. After rinsing three times with buffer solution, then, 0.25% Trypsin EDTA was used to digest cells, and a complete culture medium was added to terminate digestion. Cells were collected by centrifugation at 1000g for 10 minutes; Finally, the collected cells were resuspended in buffer solution, and the optimal excitation wavelength of 500nm and optimal emission wavelength of 525nm were selected using a fluorescence enzyme-labeled instrument for fluorescence detection.
Line 362 Genes as it is in the sentence does not appear correct; the authors should
consider changing it to singular.
Response 13:Thanks for your correction. Genes have been changed by Gene.
Conclusions
This should be improved upon to reflect on the hypothesis and results obtained.
Present study results demonstrated that NaB subdued the proliferation and increased apoptosis in BSCs. Further, NaB treatment significantly promoted mitophagy and ROS response and decreased the MMP. It was also shown that NaB might regulate apoptosis and mitophagy in BSCs by downregulating the mTOR signaling pathway. The present results indicated that NaB may be a critical metabolic molecule in muscle development, energy metabolism, and regulation of mitochondrial function.
Response 14:We sincerely appreciate the valuable comments. We have re-written this part according to the suggestion.
Conclusions
NaB is an essential nutritional element for ruminants with specific biological activities. Our study aimed to investigate the role of NaB treatment on the proliferation, apoptosis, and mitophagy of bovine skeletal muscle satellite cells (BSCs). We demonstrated that NaB inhibited the proliferation and increased apoptosis in BSCs. Further, NaB treatment significantly promoted mitophagy, increased ROS concentration, and decreased the MMP. The results also show that NaB might regulate apoptosis and mitophagy in BSCs by downregulating the mTOR signaling pathway. These findings provide novel insights into the mechanisms underlying how NaB regulates apoptosis and mitophagy and indicate that NaB may be a critical metabolic molecule in muscle development, energy metabolism, and regulation of mitochondrial function.
References
The MDPI guidelines on references were not strictly adhered to. The authors must
make all the necessary amendments in all the cited references.
Response 15:Thanks. The references have been modified according to MDPI guidelines, please refer to the manuscript for details.

Reviewer 2 Report
Please, see attached file

Redaction and editing English style should be undertaken
Author Response
Dear Editor,
We appreciate the time and effort you put into providing feedback on our manuscript, and we thank you for your insightful comments on improving our paper. your careful review has made our manuscript more clear, more comprehensive, and more readable. We have made changes to the manuscript based on your comments, and these changes are highlighted in red throughout the manuscript. The following is a point-by-point response to your comments.
Line 40. Improve redaction in this sentence, the beginning is missed.
Study showed that, NaB could inhibit the viability…
Response 1: Thanks. We have revised this sentence(Line 40).
Line 68. Improve redaction: addition of NaB to what…?
Response 2: Thanks for your careful checks. Based on your comments, we have made the corrections. Studies have shown that the addition of NaB to skeletal muscle can induce mitochondrial function by stimulating energy expenditure in mice.
Lines 103-107. Improve redaction. It appears some text is missed.
Therefore, it was preliminarily determined that the best condition to treat BSCs with 1mM NaB for 12h. Compared with the control group, after 12h of 1mM NaB treatment that the rate of total apoptotic cells was significantly increased with flow cytometry assay (P<0.01) (Fig 1d); these results indicated that NaB inhibited the proliferation and promoted apoptosis of BSCs in a time-dependent manner.
Response 3: We sincerely appreciate the valuable comments. We have made corresponding supplements in the text. Therefore, treating BSCs with 1mM NaB for 12h was preliminarily determined as the best condition. Flow cytometry assay was used to detect cell apoptosis. The results showed that, compared with the control group, after 12h treatments with 1mM NaB, early apoptotic cells were increased, while late apoptotic cells and the rate of total apoptotic cells were significantly increased (P<0.01) (Fig 1d). These results demonstrated that NaB inhibited the proliferation and promoted apoptosis of BSCs in a time-dependent manner.
Lines 132-134. Include the name of the genes in the figure legend.
Response 4: Thank you very much. We have added the name of the genes in the figure legend. Fig1. Effect of 1mM NaB on the proliferation and apoptosis of BSCs.(a) The mRNA expression of proliferation genes CCNB2, CDK1, and PCNA with 1mM NaB treatment. (b) The effect of 1mM NaB treatment on cell viability. (c) The mRNA expression of apoptosis genes Bax, Bcl-2, and p53 with 1mM NaB treatment. (d) The apoptosis rate of cells with 1mM NaB treatment by flow cytometry. Data were expressed as the mean ± standard deviation of three replicate experiments, *P < 0.05, **P < 0 .01.
Fig 2. Effect of NaB on mitophagy in BSCs. (a) Expression of mitophagy genes PINK, LC3B, and P62 with different concentrations of NaB treatment. (b) Expression of mitophagy proteins PINK, LC3B, and P62 with varying concentrations of NaB treatment. (c) Expression of mitophagy genes PINK, LC3B, and P62 with different times of NaB treatment.
Fig 3. Effect of NaB on ROS and MMP in BSCs. (a) The percentage of cells with decreased MMP. (b) The impact of 1mM NaB on cellular ROS level. (c) Ultrastructure of mitochondria under a transmission electron microscope (TEM). (A: control group; B: sodium butyrate treatment group; The red circle in B represents autophagosomes)
Fig 4. Effect of NaB on mtDNA and ATP content. (a) The expression of mtDNA copy number genes ND1 and COX1 with 1mM NaB treatment. (b) Changes of cellular ATP content with different concentrations of NaB treatment.
Fig 5. Effect of NaB on mitophagy and apoptosis through the mTOR signaling pathway. (a) The mRNA expression of key genes mTOR, EIF4EBP1, FOXO1, and AKT1 in the mTOR signaling pathway with NaB treatment. (b) The mRNA expression of key genes mTOR, EIF4EBP1, FOXO1, and AKT1 in the mTOR signaling pathway with the PP242 treatment. (c) The expression of key proteins mTOR, EIF4EBP1, FOXO1, and AKT1 in the mTOR signaling pathway with NaB and PP242 treatment, respectively. (d) The mRNA expression of apoptosis genes Bax, Bcl-2, and p53 with the PP242 treatment. (e) The mRNA expression of mitophagy genes PINK1, LC3B, and P62 with the PP242 treatment.
Line 141. What is the ROS kit? It must be compulsory to provide information regarding the method used for ROS determination (fluorimetric, colorimetric, plate reader, microscopy …). In addition, what specific ROS is detected (superoxide, hydrogen peroxide, hydroxyl radical… whatsoever) or state that ROS detection is unspecific.
Response 5: Thanks for the excellent suggestion. We have explained and provided the method as below: Determination of unspecific ROS levels by ROS assay kit: dichloro-dihydro-fluorescein diacetate (DCFH-DA, 10 μM) was treated with BSCs cells for 60 min in the dark at 37°C. After rinsing three times with buffer solution, then, 0.25% Trypsin EDTA was used to digest cells, and a complete culture medium was added to terminate digestion. Cells were collected by centrifugation at 1000g for 10 minutes; Finally, the collected cells were resuspended in buffer solution, and the optimal excitation wavelength of 500nm and optimal emission wavelength of 525nm were selected using a fluorescence enzyme-labeled instrument for fluorescence detection.
The above text has been added at Line 373-Line 381 in the manuscript.
Line 144. Bound to the above, what does ROS levels mean? It appears it should be fluorescence arbitrary units, or is it relative fluorescence? This must be clarified. Also, this is applied to figure 2b.
Response 6: ROS levels mean ROS concentration in cells, which was detected by fluorescence.
Figure 2b, you meant Figure 3b? Figure 3b was fluorescence arbitrary units. Figure 2b was the relative protein expression.
Lines 153-154. The text inside figure 2 should be unified. The font of some text is too small.
Response 7: Thanks. The text inside Figure 2 has been unified, as shown in Fig 2 of the manuscript.
Lines 171-172. Firstly, the figure legend should be placed underneath the figure panels. The name of genes should be written in the figure legend. As it is in figure 2b, there are different concentrations (0,5, 1, 2 and 5 mM). However, in line 172 only appears 1 mM. This should
be amended.
Response 8: Thanks a lot. Figure legend has been placed underneath figure panels. The name of genes has been written in the figure legend, as shown in Response 4.
1mM has been replaced by different concentrations.
Lines 200-205. Write the name of genes in figure 5 legend.
Response 9: Thanks. The name of genes in Figure 5 legend was added as shown in Response 4.
For every figure, it should be consistent with the size of font text. In some panels are too small and in other too enlarged. Keep in the middle size.
Response 10: Thanks for your suggestion. The font text size for each figure has been changed to be consistent.
Line 257. Substitute word “field”. It does not have scientific sound in this context.
Mitophagy acts as a protective scavenger to maintain intracellular environmental homeostasis by digesting different products of the organism field.
Response 11: Thanks. The word “field” was deleted in the above sentence.
Line 270. Clarify and explain “up-regulation of ROS levels,”
Response 12: “up-regulation of ROS levels” means ROS concentration increased in cells.
Lines 306-307. What is the ROS assay. Provide information. See comments mentioned for line 141.
Response 13: Thanks. We have provided information as below: Determination of unspecific ROS levels by ROS assay kit: dichloro-dihydro-fluorescein diacetate (DCFH-DA, 10 μM) was treated with BSCs cells for 60 min in the dark at 37°C. After rinsing three times with buffer solution, then, 0.25% Trypsin EDTA was used to digest cells, and a complete culture medium was added to terminate digestion. Cells were collected by centrifugation at 1000g for 10 minutes; Finally, the collected cells were resuspended in buffer solution, and the optimal excitation wavelength of 500nm and optimal emission wavelength of 525nm were selected using a fluorescence enzyme-labeled instrument for fluorescence detection.
The above text has been added at Line 373-Line 381 in the manuscript.
Lines 313-321. Improve the redaction of cell culture and provide more details. Are the cells myoblasts? or the cells had been differenciated into myotubes. Looking at the images in the supplementary material (S1), it appears that there were some myotubes. In addition, what is the skeletal muscle from which BSCs cells were isolated? How was the isolation of BSCs? This information must be stated.
Response 14: We thank you for pointing out this issue. We have re-written this part as below: Muscle tissues from the hindlimbs of the 1-month-old bovine fetus were minced into a coarse slurry and serially digested using 2% collagenase I for 1 h and in trypsinase for 30 min at 37°C. Cells were then filtered from debris, centrifuged, and cultured. For culturing, the skeletal muscle satellite cells were maintained in cell culture plates and passaged until 70% confluence was reached in DMEM-F12 complete medium supplemented with 20% FBS and antibiotics (1% penicillin-streptomycin), cells were washed twice with phosphate-buffered saline (PBS) (HyClone, USA) and digested with trypsinase for 4-5min. The digestion was terminated by adding medium immediately when the cells were observed to become rounded and suspended. The fresh cells were collected after centrifugation for use or frozen and stored in liquid nitrogen for the next experiment. All cells were grown at 37°C in a humidified atmosphere of 95% air and 5% CO2. The results of cell culture status are shown in Fig S2(Line 339-Line 350).
Lines 343-344. See comments for lines 132-134 and 144. Details regarding the fluorophore, fluorescence wavelength, fluorescence filters, exposure time for image adquisition, imaging analysis and quantification of fluorescence must be provided. Otherwise, fluorescence results might not convince. Bear in mind that to stablish fair fluorescence image analysis and comparisons, every image should be adquired with the same exposure time or be normalized properly.
Response 15: Thanks. Detailed information has been shown in above Response 13 and the method section in the manuscript.
Line 368. Use a more appropriate word “blown”. It sounds colloquial.
Response 16: Thanks. The word “blown” was replaced by “resuspend”.
Lines 384-385. Image J is an imaging analysis software. Protein content is determined by BCA. Clarify redaction and meaning.
Response 17: It has been revised, please refer to 4.10 for details.
Lines 388-394. Improve redaction of conclusions. Use a better word for “subdued”. What does mean “ROS response”. This statemen must be clarify since the results might not convince.
Response 18: We have re-written this part according to the suggestion.
Conclusions
NaB is an essential nutritional element for ruminants with specific biological activities. Our study aimed to investigate the role of NaB treatment on the proliferation, apoptosis, and mitophagy of bovine skeletal muscle satellite cells (BSCs). We demonstrated that NaB inhibited the proliferation and increased apoptosis in BSCs. Further, NaB treatment significantly promoted mitophagy, increased ROS concentration, and decreased the MMP. The results also show that NaB might regulate apoptosis and mitophagy in BSCs by downregulating the mTOR signaling pathway. These findings provide novel insights into the mechanisms underlying how NaB regulates apoptosis and mitophagy and indicate that NaB may be a critical metabolic molecule in muscle development, energy metabolism, and regulation of mitochondrial function.
Be aware that in the material methods section, there is not information regarding the immunoblotting (western-blotting) method. This must be compulsory since there are results presented in the figures.
Response 19: We think this is an excellent suggestion. We have explained the change made, including the exact location where the change can be found in the revised manuscript.
4.10 Western blot
The RIPA lysis buffer was used to separate proteins from BSCs. Then SDS- PAGE was used to separate the protein sample and then transferred to PVDF membrane and 3% albumin from bovine serum (BSA) for sealing. Then, incubate the membrane overnight at 4°C with primary antibody against PINK1 (1:1000), P62 (1:1000), LC3B (1:800), mTOR (1:1000), FOXO1 (1:800), AKT (1:1000), and GAPDH (1:500). On the second day, the membrane was incubated with the corresponding goat anti-mouse IgG or goat anti-rabbit IgG secondary antibodies at room temperature for 1 hour. After washing, membranes were detected by applying a chemiluminescent reagent and visualized using Image Lab software (Bio-Rad). Image J was employed for quantitative analysis.
Finally, it would be appreciated the discussion of other studies that assay NaB in skeletal muscle from human, mouse, and rat models, compared with the results of the present manuscript. This would be a scientific profit for the community and would reinforce this manuscript.
Response 20: As suggested by the reviewer, we have added more references to support this idea.
- Fu, Q. S.; Li, T. T.; Zhang, C.; Ma, X. T.; Meng, L. Y.; Liu, L. M.; Shao, K.; Wu, G. Z.; Zhu, X.; Zhao, X. Y., Butyrate mitigates metabolic dysfunctions via the ER alpha-AMPK pathway in muscle in OVX mice with diet-induced obesity. Cell Commun Signal 2023, 21, (1).
- Li, Y. B.; He, P. Z.; Liu, Y. H.; Qi, M. M.; Dong, W. G., Combining Sodium Butyrate With Cisplatin Increases the Apoptosis of Gastric Cancer In Vivo and In Vitro via the Mitochondrial Apoptosis Pathway. Front Pharmacol 2021, 12.
Fig S2 State of BSCs. a: State of BSCs in different proliferation periods; b: Immunofluorescence of BSCs. Left: Nuclear staining results; Right: Desmin antibody immune cell result.

Round 2
Reviewer 1 Report
IJMS-2540070-PEER-REVIEW REPORT-V2.
The authors implemented all the suggestions from the first review round. I picked up three minor errors that should be fixed.
Line 53 Change ‘plays’ to ‘play’.
Line 290 Add a ‘hyphen’ between NAB and induced.
Line 380 Check the spelling of the word ‘labeled’.

Author Response
Dear reviewer,
Thanks a lot for your careful and positive comments. We have responded all your comment point-to-point, please see below.
Referees’ comments:
The authors implemented all the suggestions from the first review round. I picked up three minor errors that should be fixed.
Line 53 Change ‘plays’ to ‘play’.
Response 1: Thanks for your careful check. The word “plays” was revised to “play”. (Line 53)
Line 290 Add a ‘hyphen’ between NAB and induced.
Response 2: We corrected it. “NaB induced” has been replaced by “NaB-induced”. (Line 290)
Line 380 Check the spelling of the word ‘labeled’.
Response 3: Thanks. We have revised ‘labeled’ to ‘labelled’. (Line 380)
